# Influence of nutrient supply on plankton microbiome biodiversity and distribution in a coastal upwelling region

Chase C. James [1,2], Andrew D. Barton[1,3], Lisa Zeigler Allen [1,2], Robert H. Lampe [1,2], Ariel Rabines[1,2], Anne Schulberg[1,2], Hong Zheng[2], Ralf Goericke[1], Kelly D. Goodwin [4] & Andrew E. Allen [1,2✉]

The ecological and oceanographic processes that drive the response of pelagic ocean microbiomes to environmental changes remain poorly understood, particularly in coastal upwelling ecosystems. Here we show that seasonal and interannual variability in coastal upwelling predicts pelagic ocean microbiome diversity and community structure in the Southern California Current region. Ribosomal RNA gene sequencing, targeting prokaryotic and eukaryotic microbes, from samples collected seasonally during 2014-2020 indicate that nitracline depth is the most robust predictor of spatial microbial community structure and biodiversity in this region. Striking ecological changes occurred due to the transition from a warm anomaly during 2014-2016, characterized by intense stratification, to cooler conditions in 2017-2018, representative of more typical upwelling conditions, with photosynthetic eukaryotes, especially diatoms, changing most strongly. The regional slope of nitracline depth exerts strong control on the relative proportion of highly diverse offshore communities and low biodiversity, but highly productive nearshore communities.

---

[1] Scripps Institution of Oceanography, University of California San Diego, 9500 Gilman Dr, La Jolla, CA 92093, United States. [2] J. Craig Venter Institute, 4120 Capricorn Lane, La Jolla, CA 92037, United States. [3] Section of Ecology, Behavior and Evolution, University of California San Diego, 9500 Gilman Dr, La Jolla, CA 92093, United States. [4] Atlantic Oceanographic and Meteorological Laboratory, (Stationed at Southwest Fisheries Science Center), 4301 Rickenbacker Cswy, Miami, FL 33149, United States. ✉email: aallen@ucsd.edu

Coastal regions disproportionately contribute to marine global primary productivity and are thus important both ecologically and economically[1]. The Southern California Current (SCC) region encompasses spatial and temporal gradients ranging from the eutrophic nearshore to the oligotrophic offshore and provides ideal conditions for quantifying variation in microbial community structure and biodiversity in response to dynamics associated with physical, chemical, and biological gradients.

Spatial patterns in marine microbial communities are strongly shaped by dispersal, environmental selection[2–5], and, on longer timescales, evolution[6]. Global-scale surveys, such as *Tara* Oceans and Malaspina[5,7–9] suggest that temperature gradients most strongly shape marine microbial community structure and biodiversity[9–11]. Other environmental conditions, such as nutrient and light availability can also provide strong bottom-up constraints in plankton communities[12,13] and are particularly important along coastal boundaries[1]. Within the SCC, coastal upwelling creates strong spatial gradients in temperature, nutrients, and light[14,15] (Supplementary Fig. 1). Previous studies have shown that phytoplankton and zooplankton communities vary along these gradients[16–18]. Furthermore, changes in seasonal nearshore upwelling are thought to drive distinct differences in phytoplankton and zooplankton assemblages across the region with variation occurring on seasonal, interannual (El Niño/La Niña), and multidecadal (Pacific Decadal Oscillation) time frames[19,20]. Within the microbial community however, the bulk of knowledge exists at a broad level across taxonomic and or functional groups, masking the effects of environmental perturbation within these broad groups and completely missing "cryptic" groups that cannot be identified with more traditional methods (such as bacterial and archaeal groups).

Metabarcoding and metagenomic datasets provide a crucial next step with which to explore the patterns and processes of marine microbial communities at a far higher resolution and in doing so, illuminate the key processes that structure the base of the marine food web. However, our current understanding of the high taxonomic resolution spatial patterns in microbial community structure and biodiversity are limited by the spatial and or temporal scale of sampling. Studies often focus on changes across space or time but rarely both[21–23]. Global datasets of marine microbiome data capture spatially extensive physical and ecological domains[5,8,24] and can identify the large environmental gradients such as temperature that appear to shape communities across large ocean basins. In contrast, investigations conducted at singular stations identify changes in the marine microbiome through time[25–28], exploring questions such as how succession within one group (such as phytoplankton) can drive changes in the overall community composition[29]. However, the biotic and abiotic mechanisms that shape biodiversity and community composition patterns often remain uncertain[4]. Combined spatial and temporal metagenomic and metabarcoding sampling of marine microbial communities is necessary to illuminate the gaps in spatially or temporally explicit microbiome studies, such as whether trends happening in one location occur elsewhere or whether observed spatial patterns are conserved or vary across time.

Here we leverage 995 microbial community composition observations from quarterly CalCOFI surveys from 2014-2020, hereafter referred to as the NOAA CalCOFI Ocean Genomics (NCOG) data. The CalCOFI surveys spans from highly productive coastal upwelling waters to oligotrophic offshore waters with NCOG sampling at both the surface and deep chlorophyll maximum (DCM, Fig. 1). With these data, we identify spatial patterns in community structure and biodiversity and highlight the environmental factors that correlate with these ecological parameters. Next, we explore how biodiversity and community structure responded to the 2014-2016 warm anomaly period, followed by the return of cooler conditions in 2017-2018. Ecological changes as a result of this shift included harmful algal blooms[30], possible poleward displacements of planktonic organisms[31], and the occurrence of novel fish species[32]. Within the SCC, it has been shown that mesoplankton communities tend to recover from other warming events (El Niño) within one year[20]. However, beyond trends in total chlorophyll[33], little is known about the response of microbial communities to the warm events in 2014-2016. Conditions were also distinct in 2019–2020 when the region experienced a smaller spring pulse of upwelling (similar to 2014–2016) that persisted from spring to early fall. To better understand the patterns and processes that shape the pelagic ocean microbiome our analyses focus on five key functional groups based on their consequential roles in marine food webs and biogeochemical cycles[34–36]: heterotrophic bacteria, cyanobacteria, archaea, and heterotrophic and photosynthetic eukaryotic protists. These functional groups are comprised of many smaller subgroups and amplicon sequence variants, or ASVs.

Within all groups, we find strong cross-shore patterns in community structure and diversity that align with gradients in nutrient supply to the surface ocean. Across both seasonal and interannual timescales, we find that the intensity of regional nutrient supply can alter cross-shore patterns in community structure varying the availability of habitat for highly productive nearshore communities. These results confirm previously observed patterns in well-studied taxonomic groups and suggest that similar environmental forcings shape the community structure and diversity of cryptic groups that were not possible to resolve through traditional techniques. Our study represents a synthesis of how both temporal and spatial environmental gradients influence microbial community assembly in a coastal upwelling biome, providing fundamental knowledge about the structure and diversity at the base of a highly productive and economically valuable ecosystem.

## Results

Across 995 samples, small subunit ribosomal RNA gene sequencing was performed on the V4-V5 region of the 16S rRNA gene for prokaryotes and the V9 region of the 18S rRNA gene for eukaryotes. Within these samples, we identified 19,204 16Sv4-5 ASVs and 34,454 18Sv9 ASVs (Supplementary Table 1). Compared to the number of 18Sv9 ASVs observed in *Tara* Oceans (207,827) or *Tara* Polar (65,655)[8], the number of ASVs found in the Southern California Current region was lower (Fig. 1e). However, of the 18Sv9 ASVs identified within NCOG, 43% were not found in either *Tara* survey, highlighting both the under-sampling of coastal ecosystems in global datasets and the added value of repeat monitoring through time towards uncovering novel regional diversity. A large proportion of the ASVs that were only found in NCOG are dinoflagellates, though many others belonged to a diverse set of taxonomic groups (Supplementary Fig. 2).

**Spatial gradients in community structure and diversity.** Nearshore to offshore gradients in community structure were an emergent property found in our self-organizing maps (SOMs; see Methods) and occurred within all five key functional groups: heterotrophic bacteria, cyanobacteria, archaea, and heterotrophic and photosynthetic eukaryotic protists (Fig. 2). SOMs are a neural-network, data reduction technique which we used to convert the highly dimensional ASV tables (995 samples × 1000 s of ASVs) into a 2-dimensional map[37]. Both surface (10 m) and deep chlorophyll

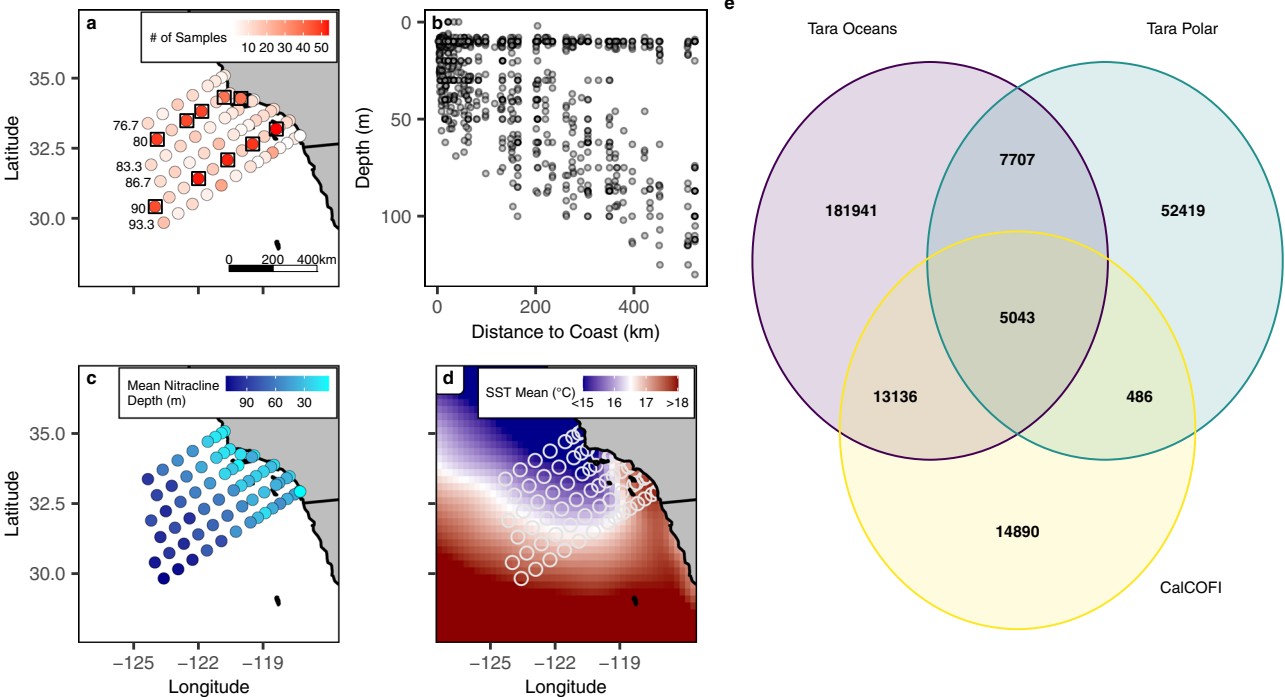

**Fig. 1 NCOG sampling and the physical environment of the Southern California Current region. a** The number of samples collected at each CalCOFI station from 2014–2020. Stations highlighted by squares are the Cardinal Stations (sampled every cruise) (**b**), location of all samples by distance to coast (*X* axis) and depth (*Y* axis) (**c**), mean nitracline depth (m) measured at each station across all seven years. **d** Mean SST (°C) from NOAA's OI SST V2 Dataset. White open circles represent the location of CalCOFI stations (**e**), overlap between NCOG 18Sv9 ASVs with those present in *Tara* Oceans and *Tara* Polar.

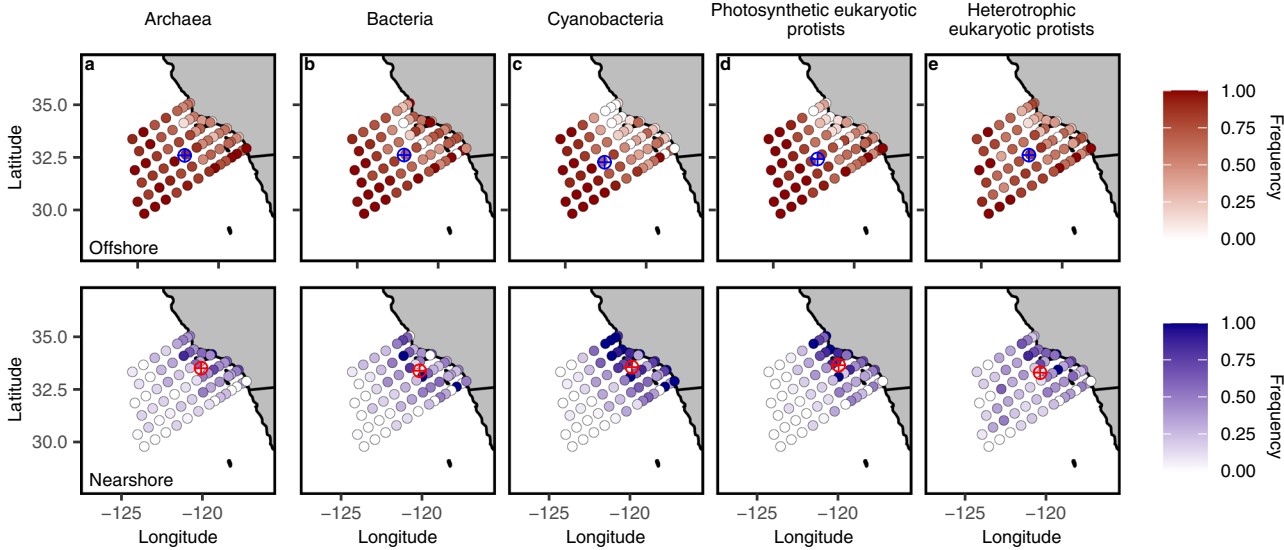

**Fig. 2 Nearshore and offshore gradients in community structure within five major microbial groups. a–e** Colors indicate the frequency that the community at each location is offshore (red, top row) or nearshore (blue, bottom row) in character. The designation of nearshore vs. offshore community is determined by the cluster whose weighted centroid is closer to the coast. The weighted centroid for each cluster is shown as a circled plus symbol in the opposite color.

maximum (DCM) samples were included in this analysis. Community clusters identified by SOMs have been subsequently labeled as "nearshore" or "offshore" based upon whether they were found more frequently in nearshore or offshore stations (weighted centroid). For the five key functional groups, these clusters aligned with waters of contrasting trophic status. On average, stations found in the northeast, nearshore corner of the sampling grid experienced mesotrophic (2.5-8 µg Chl-a L$^{-1}$) and eutrophic

conditions (>8 µg Chl-a L$^{-1}$)[38]. This contrasted strongly with the oligotrophic conditions found in most of the stations further offshore, where chlorophyll was typically low (<2.5 µg Chl-a L$^{-1}$) (Supplementary Fig. 1i).

Differences in community structure, as classified by SOM clusters, were driven by the differential relative abundance of ASVs within each of the five main groups. Within each of the five groups, there were finer-grained subgroups (e.g., SAR 11 clade

and diatoms) that exhibited differences in mean relative abundance between SOM clusters. SAR 11 ASVs were abundant in both the nearshore and offshore clusters (Supplementary Fig. 3a). However, what initially appeared to be a homogenous distribution of SAR 11 across the region was driven by three distinct SAR 11 Clade 1a ASVs: one that dominated the nearshore and two that dominated the offshore (Supplementary Data 1). One previously identified relationship within cyanobacteria[39] was observed where *Procholoroccus* ASVs had a higher relative abundance in the offshore and *Synechococcus* ASVs had a higher relative abundance in the nearshore (Supplementary Fig. 3c). Within the eukaryotic phytoplankton, diatoms were abundant in the nearshore but not the offshore SOM cluster (Supplementary Fig. 3e). Dominant nearshore diatom genera/species included: *Thalassiosira*, *Chaetoceros*, and *Pseudo-nitzchia*. In contrast, dinoflagellates dominated the offshore SOM cluster (Supplementary Fig. 3e). Dominant offshore dinoflagellates included: *Karlodinium veneficum*, *Warnowia*, and *Prorocentrum*. The ASVs that show the greatest differential abundance (>99th percentile) between nearshore and offshore clusters are provided in Supplementary Data 1.

The export rate of primary production (ef-ratio) also varied in relation to SOM clusters (Supplementary Fig. 4). Here ef-ratio is defined as new production/total production = export production/total production[40], where higher ef-ratio values indicate increased export of surface primary productivity to depth (important for carbon sequestration within the ocean). This was particularly evident in both the cyanobacteria and photosynthetic eukaryotic protists SOM clusters (Supplementary Fig. 4c, d), which both showed strong and significant relationships between the frequency with which their nearshore cluster was observed at a given station and the mean ef-ratio at that station over the seven years. The strong link between ef-ratio and proportion of nearshore and offshore communities highlights the connection between community structure and function, in this case the export of carbon from the ocean surface.

SOMs were also generated for eleven more finely resolved taxonomic groups (for a list of all groups see Supplementary Table 1). Seven out of the eleven groups showed a similar nearshore-offshore gradient in community structure, while other groups, such as *Prochlorococcus* and haptophytes showed little to no spatial patterns in community structure (Supplementary Fig. 5).

We extended the SOM analysis to examine the relationship between the frequency of observed community type (nearshore/offshore) against environmental covariates, using both the mean and coefficient of variation (coeff. var.) at each station across all seven years. In doing so, we identified the conditions across all seven years that best align with spatial patterns in the occurrence of nearshore or offshore microbial communities within the region. Coefficients of variation were included in this analysis as environmental variability is thought to promote distinct life strategies and drive population dynamics in phytoplankton species[41,42]. Nitracline depth (see Methods for definition) was a significant predictor of the nearshore-offshore gradient in community structure (lowest Akaike information criterion, AIC, Fig. 3), with the mean or coefficient of variation of nitracline depth being the most significant environmental predictor of community structure for eight out of the eleven taxonomic groups (Supplementary Fig. 6). Nitracline depth varies as the result of both abiotic and biotic factors, with upwelling bringing nutrients to the surface waters leading to a shallower nitracline and biological drawdown of nitrate within the surface ocean leading to a deepening of the nitracline. As such, nitracline depth is thought to be a critical indicator of nutrient supply into the surface ocean[43] and can be seen as both a potential driver as well

as a potential response to community changes. Mean chlorophyll *a* concentrations were also a significant predictor of the nearshore-offshore gradient in community structure (Fig. 3). However, this variable may not signify a mechanistic link, but instead reflect the ecosystem state, particularly for groups that comprise our chlorophyll *a* measurements[44].

Mean alpha (α) diversity across all ASVs, in this case calculated as the mean per station per cruise diversity, generally increased away from shore (Fig. 4a, b). For this analysis, Shannon index was used as the primary measure of diversity. The lowest mean alpha diversity was present in the northeast, nearshore subregion of the SCC and the highest mean alpha diversity was seen in the furthest offshore stations in the south. Across both surface and DCM samples (separately) we observed the same pattern of low diversity in the nearshore and high diversity offshore (Supplementary Fig. 7). Overall diversity was higher in the DCM compared to the surface, this was also true for archaea, bacteria, and cyanobacteria (Supplementary Fig. 7a–d). In contrast, autotrophic and eukaryotic protist tended to have similar levels of diversity in both the surface and DCM samples (Supplementary Fig. 7e, f). Similar increases in mean alpha diversity away from shore were found among most taxonomic subgroups (e.g., *Prochlorococcus*, SAR 11 Clade, and *Syndiniales*; Supplementary Fig. 8). However, the direction of the gradient was reversed (high diversity nearshore, low diversity offshore) for diatoms (Fig. 4d). Gamma diversity (γ; total diversity at a station over all time points) also increased away from shore (Fig. 4b), but certain groups were distinct from the pattern across all ASVs. For instance, within diatoms, mean alpha diversity was greatest nearshore, but there was little to no gradient in gamma diversity (Fig. 4d).

Nitracline depth (mean/coeff. var.) was the best predictor of spatial gradients in mean alpha diversity for all major groups except archaea (Fig. 4e) and four out of the eleven taxonomic subgroups (Supplementary Fig. 9). Three of the eleven subgroups were better predicted by the coefficient of variation in nitrate concentrations (Supplementary Fig. 9). For most groups, the relationship between nitracline depth and mean alpha diversity was positive, while, for certain groups such as diatoms, *Synechococcus*, and *Flavobacteriales*, this relationship was negative.

Previous studies have shown that diversity-productivity relationships can be unimodal[45], or vary with scale[46]. For the subset of our data where primary-productivity measurements are available (Supplementary Data 2), we found a wide variety of productivity-diversity relationships (Supplementary Fig. 10). Positive productivity-diversity relationships occurred within flavobacteria and diatoms and negative relationships occurred for the SAR 11 clades, *Prochlorococcus*, and *Syndiniales*. In some groups, the productivity-diversity relationship appeared consistent across all time periods (e.g. *Prochlorococcus*, SAR 11, *Syndiniales*), while others appeared to vary depending of the time period (Haptophytes, Chlorophytes, Dinoflagellates, Supplementary Fig. 10).

**Temporal gradients in community structure and diversity**. To better understand how community structure and diversity might be affected by temporal environmental variation, we first looked at how the environment changed over seasonal to interannual time scales in this region. Given the primary importance of nutrient supply in shaping spatial ecological gradients (Figs. 3 and 4), we focused on how coastal upwelling and nutrient availability in the surface ocean was affected across the seven-year study period.

We examined three local indices of upwelling presented by Jacox et al. (2018)[47]: Coastal Upwelling Transport Index (CUTI), Biologically Effective Upwelling Transport Index (BEUTI), and Regionally Available Nitrate (Fig. 5a–c). In the SCC, physical

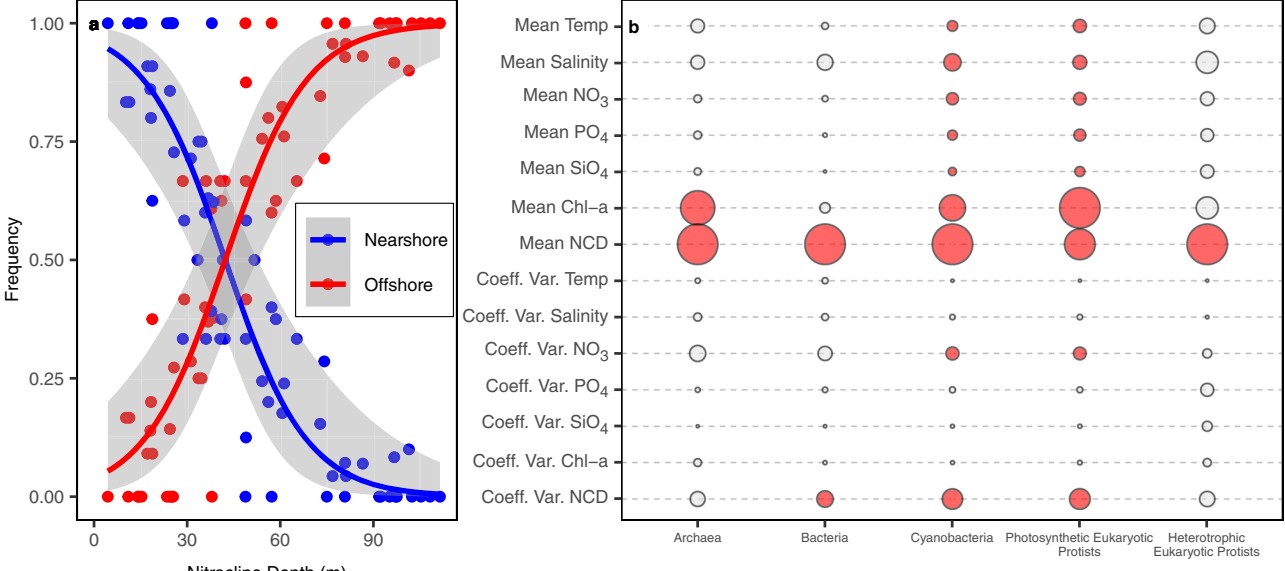

**Fig. 3 Environmental drivers of community structure. a** Example relationship between mean nitracline depth (m) and the frequency of observed community clusters ("Nearshore" in blue or "Offshore" in red) per station for cyanobacteria. Lines represent a generalized linear model with a binomial fit. Shading represents a 95% confidence interval around the model fit (**b**), relative importance of all explanatory variables (mean and coefficient of variation) used to predict the frequency of SOM clusters at a given station. As in the example (**a**), relationships were assessed via a generalized linear model with a binomial fit. Larger circles represent lower AIC values within a column; in other words, variables with larger circles are likely to be more important drivers than variables with smaller circles. Relationships that are not significant ($p > 0.05$) are colored in gray. Circles and their associated AIC values should not be compared across columns, only within columns, as AIC values are specific to each response variable. Relationships were analyzed between the frequency of observed community clusters and the mean and coefficient of variation (Coeff. Var.) of environmental variables. Environmental variables included: temperature (Temp), salinity, $NO_3$, $PO_4$, $SiO_4$, chlorophyll a (Chl-a), and nitracline depth (NCD).

upwelling (CUTI) and regionally available nitrate tend to be the lowest in late fall through winter and highest in the spring to early summer (Fig. 5a, c). While physical upwelling (CUTI) was similar throughout the years of study (Fig. 5a), the biologically effective upwelling (BEUTI) was much lower during the first three years which were affected by the 2014–2015 warm anomaly and El Niño (Fig. 5b). Upwelling in 2019-2020 was unique compared to the other years, characterized by a spring period with relatively low CUTI and BEUTI but an overall expanded upwelling season (stronger upwelling into the summer and fall relative to all other years). During the anomalously warm years 2014–2016, nitrate concentrations were relatively low in the northeast, nearshore subregion of the Southern California Current region (Fig. 5d–f). In 2014–2016, phosphate and silicate concentrations were also lower close to the coast in the northeast subregion, while concentrations of these nutrients were higher everywhere else (Supplementary Fig. 11). Mixed layer and nitracline depths across the region were similar between nearshore and offshore stations from 2014-2016—likely the result of intense stratification within the surface ocean[48] (Supplementary Fig. 11).

Interannual changes in microbial community composition across contrasting warm and cool periods were pronounced, with the largest changes occurring within eukaryotic groups (Fig. 5g–k). We compared the warm period in 2014–2016 with the relatively cool period that followed in 2017–2018, as these two periods had strongly contrasting environmental conditions. The conditions in 2019–2020, which we discuss below, were intermediate between the warm and cool phases—the offshore experienced a warm anomaly similar to 2014–2015[49], while the nearshore experienced an expanded, though moderate, upwelling season. We calculated the average community similarity (Bray-Curtis) between surface samples across the warm and cool phases for each station across our five major groups (Fig. 5g–k). Archaea, photosynthetic eukaryotic protists, and heterotrophic eukaryotic protists,

showed large shifts in community structure between the warm and cool phases (low Bray-Curtis Similarity, Fig. 5g, j, k). Cyanobacterial communities appeared to change less between the two phases than the other groups, particularly in the offshore (Fig. 5i). Changes within the samples collected at the deep chlorophyll maximum (DCM) between the warm and cool phases were less pronounced, though photosynthetic eukaryotic protist communities within the DCM were quite different between the two phases (Supplementary Fig. 12). Overall, eukaryotic groups exhibited far greater region-wide shifts in community structure between the warm and cool phases (Supplementary Fig. 13f–k). Prokaryotes, such as those ASVs assigned to the SAR 11 clade, had little to no change in community composition between the two phases (Supplementary Fig. 13a–e). Groups like *Prochlorococcus* showed almost no change in community composition in the offshore between the two phases, while simultaneously exhibiting drastic shifts in community structure in the nearshore environment (Supplementary Fig. 13a).

The 2014–2016 warm anomaly, which was localized to the upper 50 meters of the water column[48], had a clear influence on the effectiveness of physical upwelling to deliver nutrients to the surface ocean relative to 2017–2018[50] (Fig. 5, and Supplementary Fig. 11). This intense stratification may have shaped where, when, and how communities changed across the region. To test the hypothesis that temporal changes to regional stratification drove microbial community structure, we examined the relationship between the regional, cross-shore slope of nitracline depth and the proportion of samples that were identified as the nearshore (per taxonomic group via our SOMs) on a cruise-by-cruise basis. The regional slope of nitracline depth was calculated for each cruise by first flattening the sampling grid into a two-dimensional plane where the *x* axis was distance to the coast (km), and the *y* axis was the nitracline depth (m) for each station. A regional slope of the nitracline depth for each cruise was then calculated as

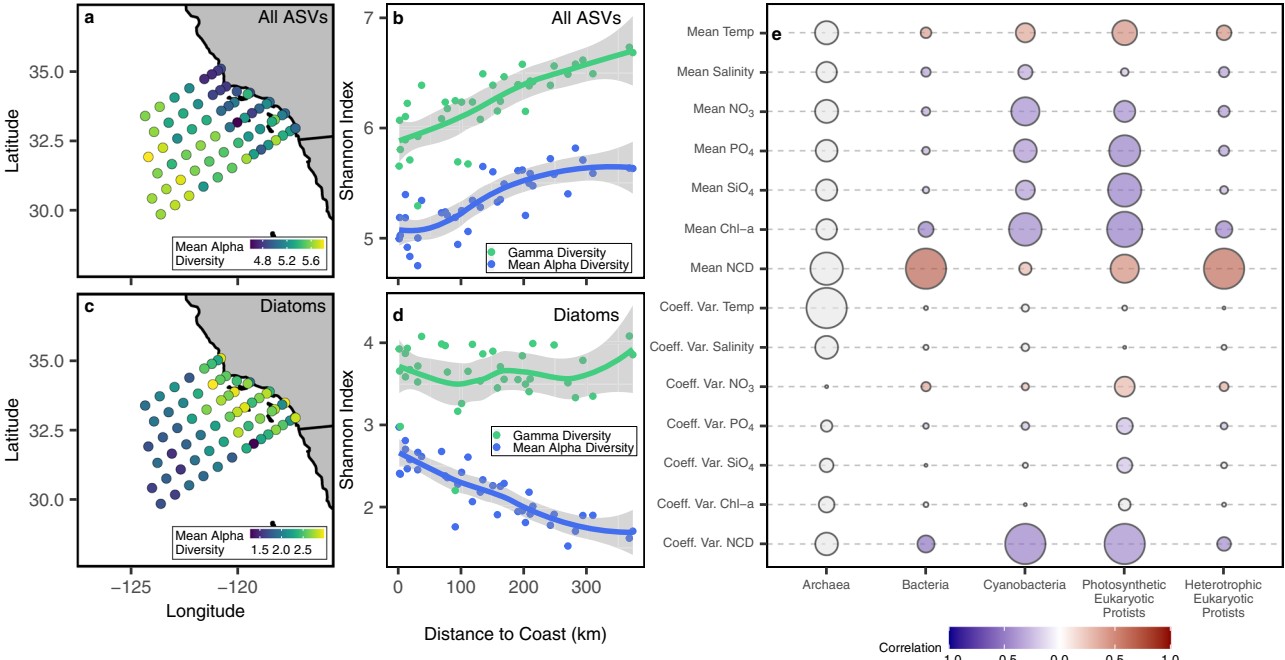

**Fig. 4 Spatial patterns and drivers of diversity. a** Map showing the mean alpha diversity for all ASVs for each station. **b** mean alpha (blue) and gamma diversity (green) per station for all ASVs as a function of distance to shore (km). Shannon index was used as the primary measure of diversity and was calculated as the mean per station per cruise for this analysis. Relationships are fit as a generalized additive model (GAM) with a 95% confidence interval. **c** map showing the mean alpha diversity for diatoms for each station. **d** mean alpha (blue) and gamma diversity (green) per station for diatoms as a function of distance to shore (km). Relationships are fit as a generalized additive model (GAM) with a 95% confidence interval (shading). **e** relative importance of all explanatory variables (mean and coefficient of variation) used to predict mean alpha diversity at a given station. Relationships between environmental variables and diversity were assessed via a generalized linear model with a gaussian fit. Larger circles represent lower AIC values within a column. Circles and their associated AIC values should not be compared across columns. Color represents the correlation coefficient between each explanatory variable and mean alpha diversity. Gray circles represent relationships that are not significant ($p > 0.05$). Relationships were analyzed between diversity and the mean and coefficient of variation (Coeff. Var.) of environmental variables. Environmental variables included: temperature (Temp), salinity, $NO_3$, $PO_4$, $SiO_4$, chlorophyll a (Chl-a), and nitracline depth (NCD).

the best linear fit through the points in this two-dimensional plane (Supplementary Fig. 14). Under normal upwelling conditions we expect the nitracline depth to be shallowest in the nearshore, coastal upwelling region, and deepest in the offshore, leading to a steep regional slope in the nitracline depth. Conversely, intense stratification of the surface ocean would promote a deeper nitracline depth in the nearshore and a shallower nitracline depth in the offshore, flattening the regional slope of nitracline depth.

We found that during the warm and cool periods, when the regional slope of nitracline depth was steeper (shallow in the nearshore and deep in the offshore), a higher proportion of samples were identified as the nearshore community type for both photosynthetic groups (cyanobacteria and photosynthetic eukaryotes) as well as bacteria. Conversely, when the regional slope of the nitracline depth was relatively flat, fewer samples were identified as nearshore (Fig. 6). Across all years (2014–2020), cruises in the spring and summer tend to have the steepest regional nitracline slopes (for an illustrative example see Fig. 6b). Fall and winter tended to have shallower regional slopes in nitracline depth and also tended to have a lower proportion of observed nearshore communities (for an illustrative example see Fig. 6a). Winter 2019 appeared to be quite distinct for this dataset, as the cruise data suggested that the region was experiencing the flattest regional slope in nitracline depth observed in all seven years, yet the proportion of nearshore communities was relatively high. However, sampling during this cruise was abnormally compressed (8-days across fewer stations) due to ship malfunction, making interpretation difficult.

Most groups tended to have a seasonal pattern in the relative dominance of nearshore/offshore communities (Supplementary Fig. 15). SAR 11 nearshore communities were more common in the spring and summer (*Flavobacteriales*, *Rhodobacterales*, metazoans showed similar trends). Other groups such as *Prochlorococcus* and diatoms showed peaks in the winter, though the presence of an increased nearshore diatom community tended to last through the spring as well (Supplementary Fig. 15). While seasonal patterns in community structure were common across all groups, the pattern was not always consistent across all years.

The 2019–2020 time period was characterized by two major anomalies, a warm, stratified layer of surface water (similar to 2014-2016) but localized to the offshore[49], and prolonged biologically effective upwelling from spring through early fall (Fig. 5b). These events combined to decrease the interseasonal variability of nutrient supply to the surface ocean within the SCC from 2019–2020. As a result, relationships between the nitracline slope and spatial extent of the nearshore communities were uncoupled in 2019–2020 (Fig. 6). This was particularly evident in diatoms and dinoflagellates, two groups that respond strongly to changes in nutrient supply[12,51], where seasonal patterns in the relative abundance of nearshore communities disappeared in 2019–2020 (Supplementary Fig. 15).

Temporal changes to mean alpha diversity occurred across both seasonal and interannual time scales. In contrast with the findings related to community structure, mean alpha diversity tended to be highest when the regional slope of nitracline depth was most flat, although, certain groups such as diatoms exhibited the reverse pattern though the relationship was not significant

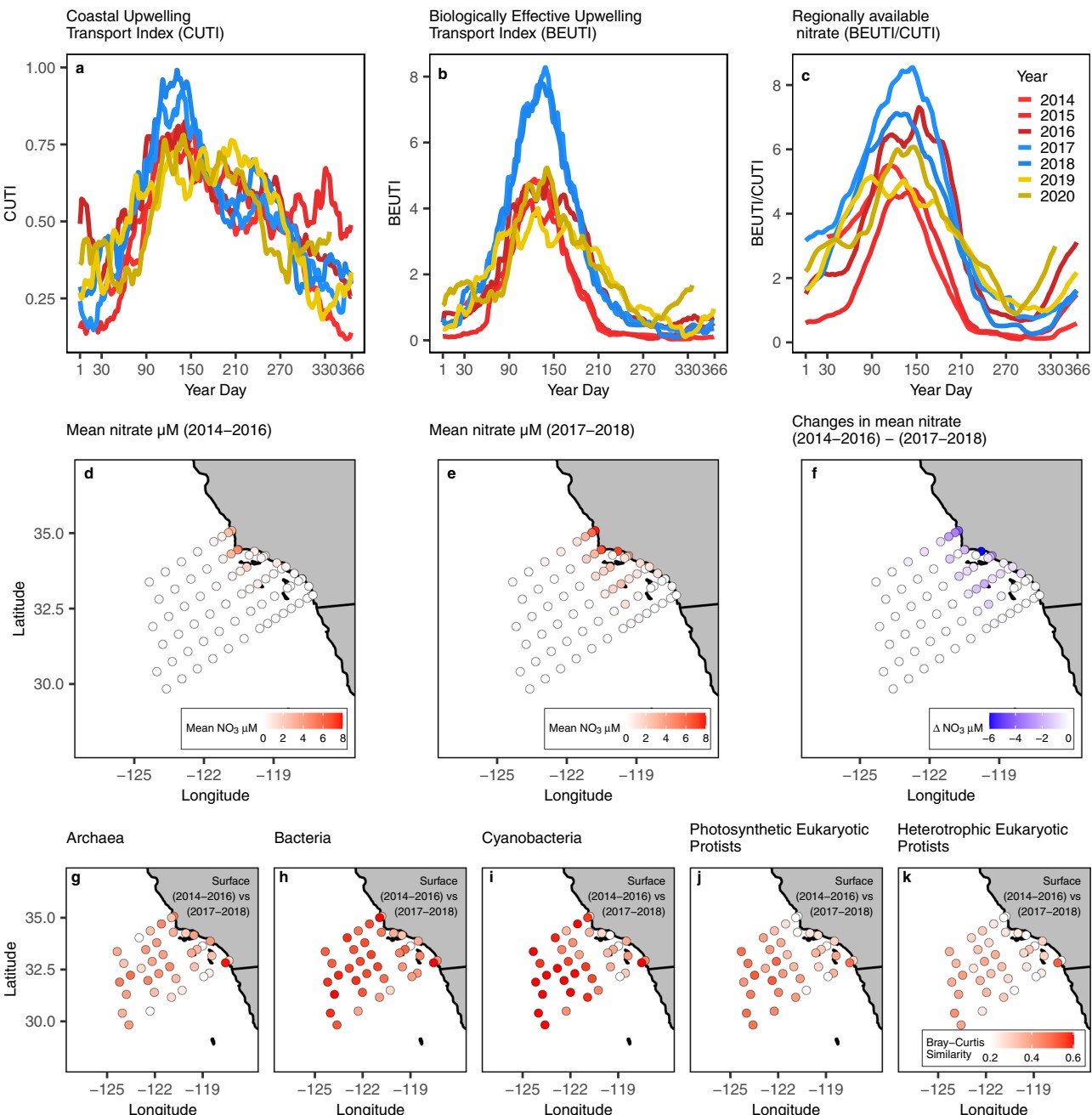

**Fig. 5 Physical and ecological changes in the region across time. a** Annual cycle of Coastal Upwelling Transport Index (CUTI). **b** Biologically Effective Upwelling Transport Index (BEUTI). **c** Regionally available nitrate for the studied time period (2014–2020). Lines are 2-month moving averages and the color palettes represent three distinct time periods, 2014–2016 (red), 2017–2018 (blue), and 2019–2020 (gold). CUTI (m$^2$ s$^{-1}$) is a regionally integrated rate of vertical volume transport. BEUTI (μM m$^{-1}$ s$^{-1}$) is an estimate of nitrate flux into the surface mixed layer. Regionally available nitrate (μM) is the concentration of nitrate at the base of the mixed layer and can be calculated by dividing BEUTI by CUTI (see Jacox et al. 2018[47] for a full explanation). **d** Mean nitrate (μM) concentrations at 10 m depth at each CalCOFI station during the warm period (2014–2016). **e** mean nitrate concentrations during the cool period. **f** The difference in nitrate concentrations between the two phases (2014–2016) – (2017–2018). **g–k** Maps of the mean Bray-Curtis similarity[92] between samples from the warm (2014–2016) and cool (2017–2018) phase for each station. Maps show surface samples for our five main groups (**g** Archaea, **h** Bacteria, **i** Cyanobacteria, **j** Photosynthetic Eukaryotic Protists, and **k** Heterotrophic Eukaryotic Protists). For DCM samples, see Supplementary Fig. 11.

(2017–2018, Supplementary Fig. 16). Like community structure, relationships between diversity and regional nitracline slope were far more frequent in the earlier years of sampling (2014–2018), when interseasonal variability in the regional nutrient supply was higher (Fig. 5b). Metazoans were the only group that showed a relationship between the regional nitracline slope and mean alpha diversity in 2019–2020 (Supplementary Fig. 16k).

## Discussion
The depth of the nitracline was a robust predictor of community structure in the SCC (Fig. 3b). In this region, the nitracline tends to be deeper in offshore waters and shallower in nearshore waters[52], creating strongly contrasting habitats. The depth of the nitracline is shaped to a great degree by the strength of upwelling; when upwelling is stronger, the nitracline is closer to the surface,

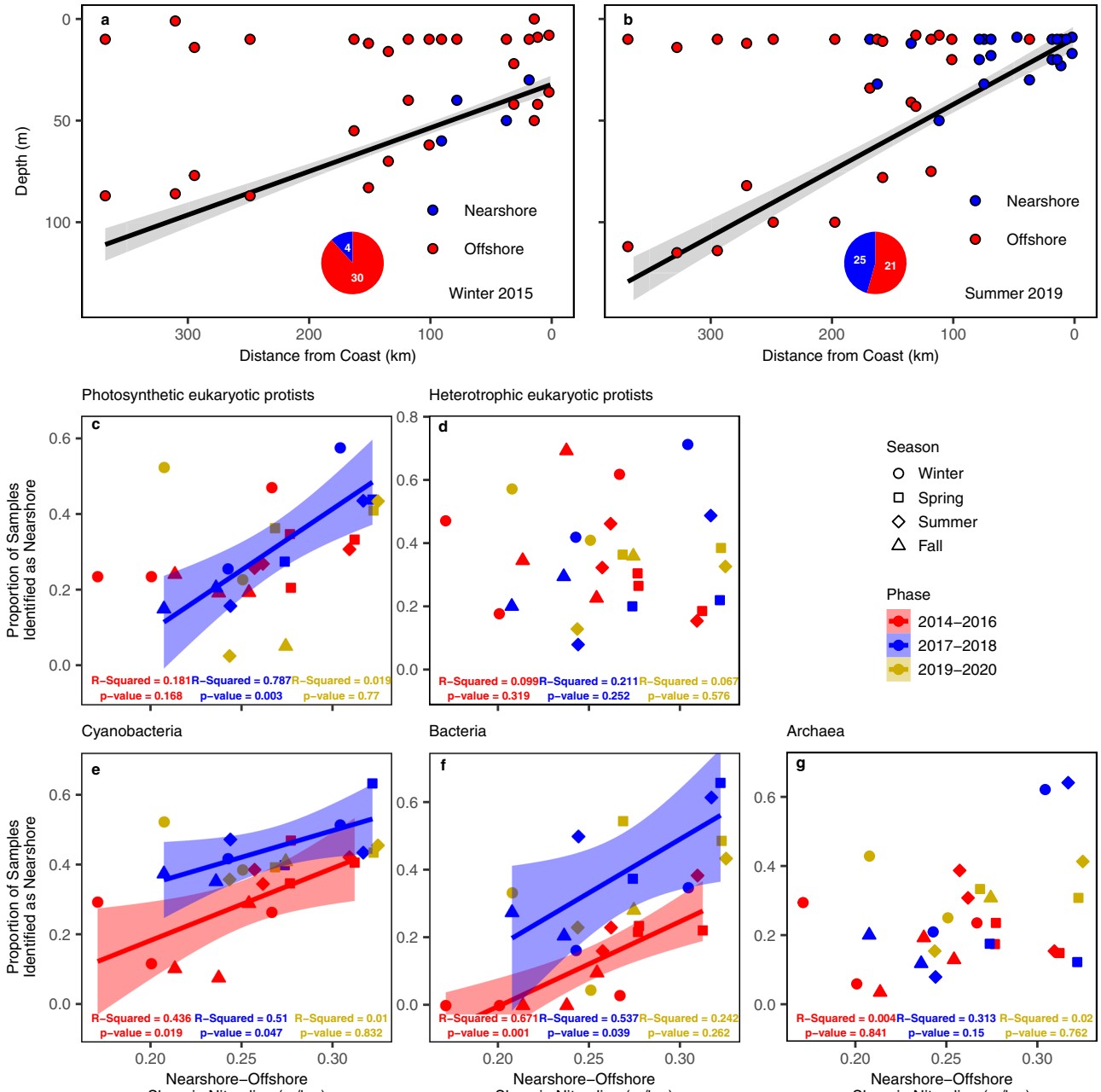

**Fig. 6 Temporal shifts in regional nitracline gradients align with relative community dominance. a** Illustrative example highlighting a cruise (Winter 2015) where the regional slope in nitracline depth is relatively low. The black line indicates the regional slope in the nitracline depth with a 95% confidence interval around the model fit (glm). Points indicate individual samples taken during this cruise. The color of the points indicates cyanobacteria communities that were identified by SOMs as either "nearshore" (blue) or "offshore" (red). **b** Illustrative example highlighting a cruise (Summer 2019) where the regional slope in nitracline depth is much greater. The black line indicates the regional slope in the nitracline depth with a 95% confidence interval around the model fit (glm). Points indicate individual samples taken during this cruise. The color of the points indicates cyanobacteria communities that were identified by SOMs as either "nearshore" (blue) or "offshore" (red). **c–g** Proportion of samples per cruise that were identified by SOMs as "nearshore" communities relative to the slope in the nitracline across the entire region. Shapes represent the different seasons during which cruises took place (circle = winter, square = spring, diamond = summer, triangle = fall) and the colors represent samples that were collected from 2014–2016 (red), 2017–2018 (blue), or 2019–2020 (gold). Data were fitted as separate linear models per phase, shading represents the 95% confidence interval around the model fit.

and the supply of nutrients to the surface is higher, if not the actual concentration of nutrients in the surface[52,53]. Nitrate limitation, as the result of variable nutrient supply, can exhibit a strong selective pressure on marine microbial communities, forcing organisms into metabolic tradeoffs in order to survive[54]. Thus, the strongly contrasting environments in the nearshore and offshore within the SCC select for very different communities.

Because nutrients are rapidly consumed by microbes in the ocean surface, the concentrations of nutrients measured represent the residual not consumed by microbes, and are in many cases not as good of a predictor of community composition when compared to the nitracline depth[15,51].

Nitracline depth was also more strongly correlated with community structure changes than temperature (Fig. 3b, Supplementary

Fig. 6). On local to global scales, nutrient availability strongly shapes primary productivity and community structure[18,55–58]. Yet in a range of recent studies, temperature has been shown to be a key correlate of global patterns of bacterial[9,10,24] (16 S) and protistan[24] (18 S) biodiversity and community structure as well as changes in the functional community composition of marine bacteria[9]. Surprisingly, these studies found little to no relationship between biodiversity, community structure, functional community composition and nitracline depth. A possible explanation is that global surveys of microbial communities have, thus far, focused their sampling effort within the open ocean, failing to capture strong coastal-open ocean physical and ecological gradients. The relative importance of environmental factors in shaping marine microbial community structure is likely to vary between regions[22] and across different spatial scales (local to global). This is likely the result of both the overall selective pressure of a variable and its relative range within the observable spatiotemporal scope of the study. Yet here within the SCC, large spatial gradients in nutrient availability, compared with temperature variability, occur with both seasonal and interannual variability, providing a testing ground to explore the selective pressure of nutrient availability in a coastal upwelling region.

Previous studies have highlighted the strong cross-shore gradients in community structure in the SCC, primarily through the use of general indices[18,59] (such as the ratio of autotrophic carbon to chlorophyll *a*) or select groups of bacteria[18], phytoplankton[15,16,18,19,60] and zooplankton[61]. The results generated from this study support and expand upon many of the findings from these previous studies. Taylor et al. 2015[18] found that the ratio of autotrophic carbon (AC) to Chl-*a* increased with increasing nitracline depth within the SCC and that the relatively low ratios of AC:Chl-*a* near the coast were a result of the dominant nearshore diatom communities which have low AC: Chl-*a* ratios. In turn, these diatom-dominated communities can lead to an "enhanced" microbial loop, with higher flows and heterotrophic bacteria standing stock biomass[62]. We find similar evidence that gradients in nitracline depth structure community composition in both phytoplankton and bacterial groups. Given the level of taxonomic resolution provided by ASVs, we were able to expand upon these prior studies to identify that these gradients also shape the taxonomic composition within groups (such as diatoms, dinoflagellates, rhodobacteria, and SAR 11 clade Supplementary Fig. 5), highlighting spatio-temporal variability in community structure at a previously inaccessible resolution. These results suggest that selection across gradients such as nutrient limitation can drive not only dominance between taxonomic groups with contrasting ecological niches and functions (diatoms vs cyanobacteria) but also drive selection within groups that are traditionally "lumped" into singular functional and or taxonomic groups (Supplementary Figs. 5, 8). Furthermore, ASVs allow for the examination of "cryptic" groups that cannot be identified through traditional approaches (microscopy, flow cytometry, chl-*a*) such as various heterotrophic bacteria (rhodobacteria, flavobacteria, SAR 11 clade) and archaea. We found that groups such as SAR 11, which are often thought to have cosmopolitan distributions, are comprised of distinct strains with varying oligotrophic to eutrophic preferences. The patterns and processes identified within this study confirm the relationship between nutrient availability microbial community structure in the SCC while further highlighting that these selective processes not only drive preferences between large functional and taxonomic groups, but also within groups.

Across most groups, mean alpha diversity was lower in the nearshore and higher offshore (Fig. 4e). The nearshore environment had relatively high nutrient concentrations and temporally variable habitats (Supplementary Fig. 1), factors which favor the competitive dominance of fast-growing, opportunistic phytoplankton such as diatoms at the expense of other species, and likely leading to lower diversity nearshore[13,45,63]. In some cases, the coefficient of variation of nitrate was a good predictor of spatial biodiversity patterns (Supplementary Fig. 9), highlighting that the nearshore environment, with its high variability and episodic pulses of nutrients, may exhibit a strong selective pressure for organisms adapted to this variable environment. An additional explanation could be that the offshore subregion of the CalCOFI grid represents a mixing zone, or ecotone, combining subtropical and coastal communities with consequently relatively high diversity[13,64–66]. The CalCOFI grid does not, however, include stations spanning deep into the subtropical North Pacific, so we cannot assess this possibility.

Diatoms presented a notable exception to the observed diversity patterns, as they showed an opposite trend in mean alpha diversity, with higher mean alpha diversity in the more productive nearshore region (Fig. 4c). While diatoms are found in subtropical waters globally, they are generally more abundant in regions and seasons with higher nutrient availability[55], and this may underpin the greater alpha diversity observed within this coastal zone. In contrast, we find no evidence of a nearshore-offshore gradient in diatom gamma diversity (Fig. 4d). This suggests that over the seven years, diatom community turnover was higher in the offshore subregion of the SCC. One possible explanation for the high overturn in diatoms but not other microbial assemblages stems from the intermittent presence of eddies and fronts in offshore waters that mediate vertical motions and nutrient supply[67–69]. Because diatoms as a group are faster-growing than other microbial groups[70], their populations respond faster to episodic pulses in nutrients than other groups. The intermittent passage of eddies and fronts in offshore waters may therefore drive an overturn of diatom ASVS while not creating a similar overturn in other groups.

Variation in the intensity of coastal upwelling across seasonal to interannual time periods controlled the relative dominance of offshore vs. nearshore community types and diversity observed within the region. During periods of strong upwelling, coastal communities were more dominant and mean alpha diversity was lower (Fig. 6, Supplementary Figs. 15, 16). Conversely, when the regional, cross-shore slope in nitracline depth was flat, most samples resembled the "offshore" community type in both structure and diversity. The 2014-2015 warm anomaly and subsequent 2015–2016 El Niño drastically reduced the extent of coastal upwelling and nutrient availability in surface waters within the region, converting nearly all available habitat into an environment that favored offshore communities. From 2014–2016, within fall and winter cruises, the majority of samples were identified as resembling an "offshore" ecotype, suggesting a drastic departure from the typical ecological gradients that exist in the region (Fig. 6). In particular, the eukaryotic assemblage changed substantially between the warm and cool phases (Fig. 5j, k). Many of the taxonomic groups showed region-wide shifts in community composition between the two phases (for example: diatoms and *Syndiniales*, Supplementary Fig. 13). 2019–2020 brought the return of the marine heatwave, though unlike 2014–2016, its effects were primarily observed offshore[49]. BEUTI measurements from the region suggest that spring upwelling for 2019–2020 had been closer to 2014–2016, however, this upwelling persisted to some degree through summer and early fall (Fig. 5b). This may have led to our observation that for certain groups such as diatoms and dinoflagellates, seasonal shifts in community structure were less pronounced (Supplementary Fig. 15). These temporal changes in the marine microbial community have implications for higher trophic levels. For example, anchovies tend to predominate in more nutrient rich coastal waters while

sardines are more abundant in oligotrophic conditions offshore[53]. Consistent with this paradigm, following the 2014–2016 warm anomaly, anchovy egg counts in Southern California reached high levels in 2017 and 2018 that had not been seen since the mid 1990s[71,72].

While previous metabarcoding studies have explored how community structure and diversity changes over time at one location[25,26,73], here we provide a comprehensive metabarcoding exploration of seasonal to interannual community variation at the regional scale. The unique lens afforded by this dataset suggests that community variability can occur across space and time, though their relative influence may vary depending on the spatial extent of temporal perturbations. We find that the depth of the nitracline is a robust predictor of both microbial community structure and biodiversity and that globally important variables such as temperature are far less predictive in the Southern California Current region. Furthermore, we found that changes in community composition could be identified not only between large functional groups, but also within groups that are often considered functionally similar. Metabarcoding also allows for the investigation of "cryptic" groups whose patterns and processes have previously been inaccessible. Across the seven years we show that changes to the spatial patterns of community structure and biodiversity coincide with seasonal and interannual changes to the steepness of cross-shore physical gradients (nitracline depth). Physical differences within the region between the warm (2014–2016) and cool (2017–2018) phases brought drastic changes in community composition, whereas reductions in the interseasonal variability of nutrient supply from 2019–2020 led to a more "static" community structure across the region. Combined, these results highlight the clear benefits of genomic surveys that sample across both space and time. Provided that there is adequate support and infrastructure to do so, future studies should be conducted in a similar manner if we are to better understand the linkages between the physical environment and microbial community structure and biodiversity.

## Methods

**Study location and sample collection.** The Southern California Current ecoregion is a component of one of the world's most productive eastern boundary currents. Productivity in the region is largely driven by seasonal upwelling—triggering the dominance of bloom forming eukaryotic phytoplankton (like diatoms) in the spring that serve as the base of a food web supporting a diverse ecosystem and many economically important fisheries[15,16,74].

Molecular and environmental data were collected on quarterly CalCOFI cruises (winter, spring, summer, and fall). At each station, seawater was collected near the surface (10 m) and the depth of the chlorophyll maximum, which varies in time and space. The chlorophyll maximum is identified on the downcast of the CTD and subsequently sampled on the upcast of the CTD. If these two depths coincided with one another then only one seawater sample was collected.

Two types of stations were sampled during this study: cardinal stations and productivity stations. Cardinal stations were sampled every cruise and occur on lines 80 (stations 55.0, 70.0, 80.0, 100.0), 81.8 (station 46.9) and 90 (stations 37.0, 53.0, 70.0, 90.0, 120.0) (Fig. 1a). Productivity stations, which measure [14]C primary production at approximately local noon were also sampled. The locations of productivity stations vary from cruise to cruise depending on where the ship is located each day at approximately local noon. Productivity stations can overlap with cardinal stations during a given cruise if the ship is located at a cardinal station at local noon.

Both molecular and environmental data were collected from a CTD rosette. Temperature and salinity were measured with a Seabird 911+ CTD. CTD salinity is validated against bottle samples which were analyzed via a Guildline Portasal Salinometer model 8410 A. Nitrate, phosphate and silicate measurements were analyzed with a QuAAtro continuous segmented flow autoanalyzer (SEAL Analytical). For chlorophyll a, seawater was filtered onto GF/F filters and then measured with the acidification method. Full methods for environmental data collection and analysis can be found at: https://calcofi.org/references/methods. At primary productivity stations, [14]C half-day incubations were started at local noon and measured as mg of carbon per m[3] per half day. Integrated primary production in the euphotic zone was then calculated as the average primary production across six light levels. For a complete procedural walkthrough of productivity incubations see: https://calcofi.org/references/methods/25-primary-productivity.html. For this study, primary productivity measurements were doubled to estimate the total production per full light day. The nitracline depth is a derived variable and is calculated as the depth where nitrate concentrations exceed or reach 1 μM via a linear interpolation based on discrete depth measurements. Metadata for all samples can be found in Supplementary Table 2.

**DNA collection and extraction.** Approximately 0.5–2 L of seawater was filtered through a 0.22 μm Sterivex-GP filter unit (MilliporeSigma, Burlington, MA, USA) for all DNA samples. Samples were immediately sealed with a sterile luer-lock plug and hematocrit sealant, wrapped in aluminum foil, and flash frozen in liquid nitrogen. DNA was extracted with the NucleoMag Plant Kit for DNA purification (Macherey-Nagel, Düren, Germany) on an epMotion 5057TMX (Eppendorf, Hamburg, Germany) as described here: https://doi.org/10.17504/protocols.io. bc2hiyb6. DNA was assessed on a 1.8% agarose gel after extraction.

**Amplicon sequencing and analysis.** Amplicon libraries targeting the V4-V5 region of the 16 S rRNA gene and V9 region of the 18 S rRNA gene were generated as described here: https://www.protocols.io/view/amplicon-library-preparation-bmuck6sw. Briefly, DNA was amplified via a one-step PCR using the TruFi DNA Polymerase PCR kit (Azura, Raynham, MA, USA). For 16 S, the 515 F (GTGYCAGCMGCCGCGGTAA) and 926 R (CCGYCAATTYMTTTRAGTTT) primer set was used[75]. For 18 S, the 1389 F (TTGTACACACCGCCC) and 1510 R (CCTTCYGCAGGTTCACCTAC) primer set was used[76]. Each reaction was performed with an initial denaturing step at 95 °C for 1 min followed by 30 cycles of 95 °C for 15 sec, 56 °C for 15 sec, and 72 °C for 30 sec. Custom mock communities[75] were included in the sequencing runs (Supplementary Fig. 17). 2.5 μL of each PCR reaction was ran on a 1.8% agarose gel to confirm amplification. PCR products were purified using Beckman Coulter AMPure XP beads following the standard 1x PCR clean-up protocol. PCR quantification was performed in duplicate using Invitrogen Quant-iT PicoGreen dsDNA Assay kit. Samples were then pooled in equal proportions into seven pools for the 16S data and five pools for the 18Sv9 data followed by another 0.8x AMPure XP bead purification. Pools was evaluated on an Agilent 2200 TapeStation and quantified with Qubit HS dsDNA. Each pool was sequenced at the University of California, Davis Sequencing Core on a single Illumina MiSeq lane (2 × 300 bp for 16 S, 2 × 150 bp for 18 S) with a 15% PhiX spike-in. For the 2014–2016 data, the 18Sv9 pool was sequenced on an Illumina NextSeq (2 × 50 bp).

Amplicons were analyzed with QIIME2 v2019.104[77]. Briefly, demultiplexed paired-end reads were trimmed to remove adapter and primer sequences with cutadapt[78]. Trimmed reads were then denoised with DADA2 to produce amplicon sequence variants (ASVs)[79]. Each pool was denoised with DADA2 individually to account for different error profiles in each run. Taxonomic annotation of ASVs was conducted with the q2-feature-classifier classify-sklearn naïve-bayes classifier[80,81] against SILVA (Release 138)[82] for 16S amplicons or PR[2] v4.13.0[83] for 18Sv9 amplicons.

*Tara* Oceans and *Tara* Polar data were downloaded from the European Nucleotide Archive under the project accessions PRJEB6610 [https://www.ebi.ac. uk/ena/browser/view/PRJEB6610] and PRJEB9737 [https://www.ebi.ac.uk/ena/browser/view/PRJEB9737]. Raw sequences were analyzed in the QIIME2 environment with DADA2 as described above. As run information was not available, each sample was analyzed with DADA2 individually; however, on average each sample contains enough reads to accurately estimate the error rates (>1 million reads).

For this study, we rarefied our libraries to 17,000 reads, maintaining 99% of our samples (11 were removed due to small library sizes). While there have been arguments on either side concerning rarefaction in microbiome datasets[84–86], we believe that the wide variability in our library sizes, ranging from thousands of reads to hundreds of thousands of reads, justifies our decision to rarefy—large differences in library size can drastically alter biodiversity estimates[86].

**Biodiversity metrics.** The Shannon Index was used in our measures of both alpha and gamma diversity. Mean alpha diversity was calculated per station (Fig. 4, Supplementary Fig. 9) or per cruise (Fig. 6, Supplementary Fig. 16). Gamma diversity was calculated by summing together all observed reads per station before calculating a Shannon Index to get the total gamma diversity per station across all seven years of sampling (Fig. 4). Beta diversity was calculated as a Bray-Curtis Similarity (Fig. 5, Supplementary Figs. 12, 13). Both Shannon Index and Bray-Curtis similarity were calculated using the *vegan* package in R[87].

**Self-organizing maps (SOMs).** Self-Organizing Maps (SOMs) are a data reduction technique capable of reducing highly variable data into a two-dimensional map while retaining properties of the original highly dimensional data. Consequently, SOMs are suitable for identifying distinct ecological communities with amplicon data, as they can reduce the complexity of tens of thousands of unique species (ASVs) to a small set of discrete communities[88]. For these data we generated the SOMs on a 6 × 6 neuronal map using the SOMbrero package in R[89]. SOMs included all 984 individual samples. For each taxonomic group, once a SOM was generated, hierarchical clustering was used to cluster neurons (nodes of the map) together, identifying the two most distinct community clusters present on the

maps (see Supplementary Table 1 for a list of taxonomic groups). See Fig. 2 and Supplementary Fig. 5 for station maps representing the relative dominance between the two SOM clusters for all taxonomic groups.

**Generalized linear models (GLM)**. Generalized linear models (GLMs) were used to test the relative importance of environmental conditions on plankton community structure in the California Current. For the first set of models (Fig. 3 and Supplementary Fig. 6), the response variable was the frequency at which a specific community (nearshore or offshore), as defined by the SOMs, was found at a given station. A binomial fit was used as the range of possible values was between 0 and 1. For the second set of models (Fig. 4 and Supplementary Fig. 9), the response variable was mean alpha diversity. In this case, the fit was normal as the distribution of mean alpha diversity values was close to normal. GLM's only considered stations with at least four data points (one year). Single parameter models were compared to one another using the Akaike Information Criterion (AIC) to identify the most suitable model[90].

**Generalized additive models (GAM)**. Generalized additive models were used to fit Shannon index-distance to coast (Fig. 4a, d) and productivity-diversity relationships (Supplementary Fig. 10). Here, GAMs were used as they provide a flexible and simple means of identifying relationships between variables without the need to specify a specific type of relationship (linear, exponential, logistic) per fit.

**Reporting summary**. Further information on research design is available in the Nature Research Reporting Summary linked to this article.

## Data availability
The 16 S rDNA raw reads have been deposited at NCBI under Bioproject IDs PRJNA555783, PRJNA665326 and PRJNA804265 and Biosample accession nos. SAMN25705811-SAMN25706151, SAMN16250568-SAMN16251083, and SAMN25756929-SAMN25757078 and for the 2014–2016, 2017–2019, and 2020 periods respectively. The 18 S rDNA raw reads have been deposited at NCBI under Bioproject IDs PRJNA555783, PRJNA665326, and PRJNA804265 and Biosample accession nos. SAMN25710021-SAMN25710361, SAMN16251281-SAMN16251796, and SAMN25757352-SAMN25757501 for the 2014–2016, 2017–2019, and 2020 periods respectively. *Tara* Oceans and *Tara* Polar 18Sv9 sequences can be found at the European Nucleotide Archive under the project accession IDs PRJEB6610 and PRJEB9737 respectively. Associated sample metadata are provided in the Supplementary Data 2 file.

## Code availability
The code for this study is located at https://github.com/ChaseCJames/NCOG_Spatial_Environ[91]. https://doi.org/10.5281/zenodo.6359865.

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

## Acknowledgements

C.C.J. acknowledges graduate student support by Scripps Institution of Oceanography. This study was supported by National Science Foundation, California Current Ecosystem Long Term Ecological Research Grants, CCE-LTER Phase II (NSF-OCE-1026607 and NSF-OCE-1637632), NSF-OCE-1756884, NOAA (NOAA OAR Omics, CIMEC NA15OAR4320071 and ECOHAB NA19NOS4780181), and Gordon and Betty Moore Foundation grants GBMF3828 to AEA. We would like to acknowledge former CalCOFI director David M. Checkley and Margot Bohan from the NOAA Office of Ocean Exploration and Research (OER) for their vision and guidance during the initial

phase of the NCOG program and current CalCOFI director Brice X. Semmens for his continued support. We are also especially grateful to California Current Ecosystem, Long Term Ecological Research (CCE-LTER) and CalCOFI project and team members and crew who have assisted with the NCOG program 2014-present.

## Author contributions
L.Z.A., R.G., R.D.G., K.D.G., and A.E.A. designed and implemented NCOG within the CalCOFI program. AS conducted most of the NCOG sampling aboard CalCOFI cruises. R.H.L., A.R., and H.Z. performed DNA purifications and constructed metabarcoding amplicon libraries. R.H.L. generated the ASV tables used for analyses. C.C.J., A.D.B., A.E.A., L.Z.A, and R.D.G. developed the core analyses for the study. C.C.J. ran the statistical analyses, generated figures, and was the primary author of the manuscript. All co-authors contributed to editing of the manuscript.

## Competing interests
The authors declare no competing interests.
