## [Peer Review File · Nature Communications]

Influence of nutrient supply on plankton microbiome biodiversity and distribution in a coastal upwelling regionEditorial Note: This manuscript has been previously reviewed at another journal that is not operating a transparent peer review scheme. This document only contains reviewer comments and rebuttal letters for versions considered at Nature Communications.

Reviewers' Comments:

Reviewer #1:

Remarks to the Author:

The current manuscript describes microbial diversity in a Southern California Current region as observed between 2014-2020. This manuscript was previously submitted to Nature Microbiology and has been transferred to Nature Communications. I was a reviewer of the earlier submission. I have now evaluated the authors' responses to previous reviewer comments and the revised manuscript. I believe the authors have addressed my earlier concerns and have prepared an interesting paper for Nature Communications. I only have a few minor suggestions:

(line 67) I do not understand the beginning of this sentence – "To address these inquires our analyses focus". please rephrase

(line 77) should this sentence end with "...and the V9 region fo the 18S rRNA gene for eukaryotes"?

(line 126) change to "and can be seen as"

(line 129) change to "but instead reflect the ecosystem"

(line 197-198) change to "Prokaryotes, such as those AVSs"

(line 209) change to "identified as nearshore"

(line 252) change to "only group that showed"

(line 342) change to "space and time"

(line 349) change to "cool (2017-2018) phases"

(line 350-351) change to "2019-2020 led to a more 'static' community structure across the region. Combined..."

(line 352) change to "...both space and time. Provided..."

(line 358) change to "one of the world's most productive eastern boundary currents."

(line 381) change to "incubations were conducted at local noon and measured as mg of carbon..." and you should probably change the wording here an elsewhere to state that "half-day incubations were started at local noon" rather than "conducted at local noon"

Figures: Why are different 'dot' sizes used for different figures (e.g., Supplementary figures 6 & 7)? Perhaps this should be standardized.

Figure 1: perhaps add to figure 1d open circles indicating sampling station locations to be consistent

with figure 1a and 1c

Reviewer #3:

Remarks to the Author:

The authors have substantially improved the manuscript over the last version, especially related to the technical issues associated with different filter types and including an analysis of the 2019-2020 data. This work represents a substantial effort on the part of the authors to collect, sequence and analyze microbiome data from a large number of samples. It provides an in-depth look into the spatial and temporal changes to microbial community structure, focusing on important groups (e.g. protists, eukaryotic algae, cyanobacteria, heterotrophs) and is a nice integration of the sequence-based biological results with the physical and chemical structure of the ocean in this region. Their major results include the differentiation of near-shore and off-shore microbial communities, that nitracline depth best explains the proportion of near/off-shore members within the community, and how community composition (marked by fraction of near-shore/off-shore community members) changes between periods of warm/cold anomalies related to the changes in nitracline slope.

The main shortfalls of this manuscript are that i.) it does not properly frame their results in the context of previous studies of microbial community structure and diversity in the coastal ocean and ii.) it does little to translate the changes in diversity or near/off-shore groups into meaningful changes in community function.

i.) The lack of context for their results based on what has been done in other work is highlighted by the authors' response discussion of previous work at this site (Reviewer #3 comment 5). Rather than discuss the similarities between previous work and what others found, they make a general statement that those analyses were "restricted". But the main point is that they found a relationship with nitracline depth and autotrophic carbon: chlorophyll a ratios which the authors had attributed to differences in diatoms in near/offshore sites. This should be highlighted in their work, and the authors should build on this work in their discussion as the starting point to describe what additional insight this analysis brought.

In the author's response, they acknowledge their analysis "confirms some existing ideas about how marine microbial communities are distributed." The authors also try to explain why their analysis represents a "fundamental step forward", but use language that is vague and not particularly compelling, such as "describe the patterns and processes that shape community structure and biodiversity across a far greater diversity of marine microbial groups." Just describing patterns, especially those that are similar to patterns observed previously, is not a fundamental step forward, even though it has never "been done before at this combined temporal and spatial scale". The authors should be able to more clearly articulate the important findings of this work compared to what was known before the study and how knowledge has advanced.

The authors should provide better context in the introduction about what is known already about the relationship of microbial diversity with upwelling regions or near-shore/off-shore gradients and changes over time due to warm/cool anomalies. This will provide context for their results. Specifically, in the second paragraph, the authors could summarize studies of microbial diversity that have been done in the coastal ocean and the "rare" studies that have been conducted in both space and time to clearly demonstrate what additional insight was gained by combining space and time, not just that insight was gained.

ii.) The characterization of communities as nearshore/off-shore types is not particularly meaningful, as

their position from shore doesn't convey useful information about community function. Could these different communities also be called oligotrophic (for off-shore) and eutrophic (for nearshore) communities given the statement in the abstract describing the transition as "eutrophic nearshore to the oligotrophic offshore"? If so, this would be much more insightful than describing these communities as nearshore/offshore, which doesn't have much biological meaning other than position from the coast. This then highlights the idea that typical eutrophic organisms are found with more upwelling, which again either supports what is known about upwelling regions, or a better description needs to be provided about how it is similar/different from what is known from previous work.

Along these lines, could the authors also create a schematic to illustrate the relationship between the nearshore/offshore nitracline slope, up-welling and the community composition to facilitate understanding of how this change influences community composition and function? It might be particularly compelling if the authors adopt the oligotrophic/eutrophic communities nomenclature and show the steeper nitracline slope created from stronger nutrient upwelling leads to a larger proportion of eutrophic(nearshore) community members overall. This would make it easier for readers to understand the relationship between these different components of their work. While Extended data Fig. 13 is helpful to outline the relationships between BEUTI, nitracline slope and spatial relationship with distance from shore, a schematic about how this impacts the microbial community would be helpful.

Fig. 3 How does the nearshore community make up 100% of the sample when the nitracline depth is largest (90m) since nitracline depth increases with distance from shore? Why did this figure change completely from the original form? This is confusing and should be explained in either the legend or the text.

Response to Reviewers (2nd Round)

We thank the reviewers again for their detailed comments and suggestions. Below, we carefully address each reviewer's comment in sequence. As before, our responses to the reviewers are in **bold** and reviewer comments are in regular font

Reviewer #1 (Remarks to the Author):

The current manuscript describes microbial diversity in a Southern California Current region as observed between 2014-2020. This manuscript was previously submitted to Nature Microbiology and has been transferred to Nature Communications. I was a reviewer of the earlier submission. I have now evaluated the authors' responses to previous reviewer comments and the revised manuscript. I believe the authors have addressed my earlier concerns and have prepared an interesting paper for Nature Communications. I only have a few minor suggestions:

We thank the reviewer for the time and effort reviewing this manuscript. Their suggestions and comments have greatly improved the manuscript over these two sets of revisions. We have incorporated all their minor suggestions below.

(line 67) I do not understand the beginning of this sentence – “To address these inquires our analyses focus”. please rephrase

We have rephrased this sentence (now Line 63) with the following updated text, “To better understand the patterns and processes that shape the pelagic ocean microbiome our analyses focus on five key functional groups based on their consequential roles in marine food webs and biogeochemical cycles: ... ”.

(line 77) should this sentence end with “...and the V9 region fo the 18S rRNA gene for eukaryotes”?

Thank you for noticing this, we have updated this in the text (now Line 72) “Across 995 samples, small subunit ribosomal RNA gene sequencing was performed on the V4-V5 region of the 16S rRNA gene for prokaryotes and the V9 region of the 18S rRNA gene for eukaryotes”.

(line 126) change to “and can be seen as”

This has been updated in the text (now Line 138):

“As such, nitracline depth is thought to be a critical indicator of nutrient supply into the surface ocean and can be seen as both a potential driver as well as a potential response to community changes.”

(line 129) change to “but instead reflect the ecosystem”

This has been updated in the text (now Line 142):

“Mean chlorophyll a concentrations were also a significant predictor of the nearshore-offshore gradient in community structure (Fig. 3). However, this variable may not signify a mechanistic link, but instead reflect the ecosystem state, particularly for groups that comprise our chlorophyll a measurements.”

(line 197-198) change to “Prokaryotes, such as those AVSs”

This has been updated in the text (now Lines 210-212):

“Prokaryotes, such as those ASVs assigned to the SAR 11 clade, had little to no change in community composition between the two phases”

(line 209) change to “identified as nearshore”

This has been updated in the text (Now Lines 235-236):

“Conversely, when the regional slope of the nitracline depth was relatively flat, fewer samples were identified as nearshore”

(line 252) change to “only group that showed”

This has been updated in the text (Now Lines 266-267):

“Metazoans were the only group that showed a relationship between the regional nitracline slope and mean alpha diversity in 2019-2020”

(line 342) change to “space and time”

This has been updated in the text (now Lines 371-373):

“The unique lens afforded by this dataset suggests that community variability can occur across space and time, though their relative influence may vary depending on the spatial extent of temporal perturbations”

(line 349) change to “cool (2017-2018) phases”

This has been updated in the text (now Lines 381-382):

“Physical differences within the region between the warm (2014-2016) and cool (2017-2018) phases brought drastic changes in community composition...”

(line 350-351) change to “2019-2020 led to a more ‘static’ community structure across the region. Combined...”

This has been updated in the text (now Lines 382-384):

“...whereas reductions in the interseasonal variability of nutrient supply from 2019-2020 led to a more “static” community structure across the region.”

(line 352) change to “...both space and time. Provided...”

This has been updated in the text (now Lines 384-385):

“...these results highlight the clear benefits of genomic surveys that sample across both space and time. Provided that...”

(line 358) change to “one of the world’s most productive eastern boundary currents.”

This has been updated in the text (now Lines 390-391):

“The Southern California Current ecoregion is a component of one of the world’s most productive eastern boundary currents.”

(line 381) change to “incubations were conducted at local noon and measured as mg of carbon...” and you should probably change the wording here an elsewhere to state that “half-day incubations were started at local noon” rather than “conducted at local noon”

Thanks for this suggestion the text has been updated to reflect this:

“At primary productivity stations, ¹⁴C half-day incubations were started at local noon and measured as mg of carbon per m³ per half day.” Lines 412-413

Figures: Why are different ‘dot’ sizes used for different figures (e.g., Supplementary figures 6 & 7)? Perhaps this should be standardized.

Point sizing for plots is dependent on the size of the subfigure so there is some variability in point size per plot. However, for map plots (Such as Supplementary Figures 6 & 7) we have resized points in these figures to be consistent across all figures. Thank you for your suggestion.

Figure 1: perhaps add to figure 1d open circles indicating sampling station locations to be consistent with figure 1a and 1c

Thank you for this suggestion we have updated the figure to include open circles in Fig. 1d.

Reviewer #3 (Remarks to the Author):

The authors have substantially improved the manuscript over the last version, especially related to the technical issues associated with different filter types and including an analysis of the 2019-2020 data. This work represents a substantial effort on the part of the authors to collect, sequence and analyze microbiome data from a large number of samples. It provides an in-depth look into the spatial and temporal changes to microbial community structure, focusing on important groups (e.g. protists, eukaryotic algae, cyanobacteria, heterotrophs) and is a nice integration of the sequence-based biological results with the physical and chemical structure of the ocean in this region. Their major results include the differentiation of near-shore and off-shore microbial communities, that nitracline depth best explains the proportion of near/off-shore members within the community, and how community composition (marked by fraction of near-shore/off-shore community members) changes between periods of warm/cold anomalies related to the changes in nitracline slope.

The main shortfalls of this manuscript are that i.) it does not properly frame their results in the context of previous studies of microbial community structure and diversity in the coastal ocean and ii.) it does little to translate the changes in diversity or near/off-shore groups into meaningful changes in community function.

i.) The lack of context for their results based on what has been done in other work is highlighted by the authors' response discussion of previous work at this site (Reviewer #3 comment 5). Rather than discuss the similarities between previous work and what others found, they make a general statement that those analyses were "restricted". But the main point is that they found a relationship with nitracline depth and autotrophic carbon: chlorophyll a ratios which the authors had attributed to differences in diatoms in near/offshore sites. This should be highlighted in their work, and the authors should build on this work in their discussion as the starting point to describe what additional insight this analysis brought.

In the author's response, they acknowledge their analysis "confirms some existing ideas about how marine microbial communities are distributed." The authors also try to explain why their analysis represents a "fundamental step forward", but use language that is vague and not particularly compelling, such as "describe the patterns and processes that shape community structure and biodiversity across a far greater diversity of marine microbial groups." Just describing patterns, especially those that are similar to patterns observed previously, is not a fundamental step forward, even though it has never "been done before at this combined temporal and spatial scale". The authors should be able to more clearly articulate the important findings of this work compared to what was known before the study and how knowledge has advanced.

We thank the reviewer for their comments and have reworked relevant sections of the discussion to comment on the similarities between previous work and what we have found in our study. We highlight that our results align with studies such as Taylor et al. 2015 which found a similar relationship between community

composition (as defined by the ratio of autotrophic carbon to chlorophyll-a) and nitracline depth within the region. We also comment on how this study builds upon this foundation of knowledge. In particular, we suggest that the high resolution afforded by ASVs allows for a better understanding of how environmental conditions shape community composition within taxonomic/functional groups. This can be seen as a fundamental step forward as we find that 1) within groups that are often labeled and functionally similar (such as diatoms) there still exists strong selective pressures of nutrient availability on community composition and that 2) we are able to explore “cryptic” groups, such as heterotrophic bacteria and Archaea, that cannot be identified through more traditional approaches.

The full changes to the text have been included below (Lines 297-321):

“Previous studies have highlighted the strong cross-shore gradients in community structure in the SCC, primarily through the use of general indices^{19,61} (such as the ratio of autotrophic carbon to chlorophyll a) or select groups of bacteria¹⁹, phytoplankton^{16,17,19,20,62} and zooplankton⁶³. The results generated from this study support and expand upon many of the findings from these previous studies. Taylor et al. 2015¹⁹ found that the ratio of autotrophic carbon (AC) to Chl-a increased with increasing nitracline depth within the SCC and that the relatively low ratios of AC:Chl-a near the coast were a result of the dominant nearshore diatom communities which have low AC: Chl-a ratios. In turn, these diatom-dominated communities can lead to an “enhanced” microbial loop, with higher flows and heterotrophic bacteria standing stock biomass⁶⁴. We find similar evidence that gradients in nitracline depth structure community composition in both phytoplankton and bacterial groups. Given the level of taxonomic resolution provided by ASVs, we were able to expand upon these prior studies to identify that these gradients also shape the taxonomic composition within groups (such as diatoms, dinoflagellates, rhodobacteria, and SAR 11 clade Supplementary Data Fig. 5), highlighting spatio-temporal variability in community structure at a previously inaccessible resolution. These results suggest that selection across gradients such as nutrient limitation can drive not only dominance between taxonomic groups with contrasting ecological niches and functions (diatoms vs cyanobacteria) but also drive selection within groups that are traditionally “lumped” into singular functional and or taxonomic groups (Supplementary Data Fig. 5, Supplementary Data Fig. 8). Furthermore, ASVs allow for the examination of “cryptic” groups that cannot be identified through traditional approaches (microscopy, flow cytometry, chl-a) such as various heterotrophic bacteria (rhodobacteria, flavobacteria, SAR 11 clade) and Archaea. We found that groups such as SAR 11, which are often thought to have cosmopolitan distributions, are comprised of distinct strains with varying oligotrophic to eutrophic preferences. The patterns and processes identified within this study confirm the relationship between nutrient availability microbial community structure in the SCC while further highlighting that these selective processes not only drive preferences between large functional and taxonomic groups, but also within groups.”

Lines 375-378

“Furthermore, we found that changes in community composition can be found not only between large functional groups, but also within groups that are often considered functionally similar. Metabarcoding also allows for the investigation of “cryptic” groups whose patterns and processes have previously been inaccessible.”

The authors should provide better context in the introduction about what is known already about the relationship of microbial diversity with upwelling regions or near-shore/off-shore gradients and changes over time due to warm/cool anomalies. This will provide context for their results. Specifically, in the second paragraph, the authors could summarize studies of microbial diversity that have been done in the coastal ocean and the “rare” studies that have been conducted in both space and time to clearly demonstrate what additional insight was gained by combining space and time, not just that insight was gained.

We thank the reviewer for their comments and have reworked both the second and third paragraphs of the introduction to address their comments.

In the second paragraph (Lines 21-34) we better summarize previously identified relationships between community structure and cross shore gradients in nutrient supply within the SCC. We highlight studies that have shown similar spatial patterns as well as studies that have documented the relationship between nearshore community abundance and interannual variability in upwelling.

We then identify that a current “missing” component of these results is the ability to discern how environmental gradients shape communities at the sub-group level, including “cryptic” groups that have either been ignored or too difficult to identify. Following lines 21-31, we expand upon the insights generated from both global and station-based metagenomic and metabarcoding sampling. We contextualize the NCOG sampling regime (a regional grid sampled quarterly across seven years) as a fundamental step forward in sampling effort within the marine environmental genomic landscape.

Full text for these paragraphs is shown below (Lines 21-49):

“Spatial patterns in marine microbial communities are strongly shaped by dispersal, environmental selection²⁻⁵, and, on longer timescales, evolution⁶. Global-scale surveys, such as Tara Oceans and Malaspina^{5,7-9} suggest that temperature gradients most strongly shape marine microbial community structure and biodiversity⁹⁻¹¹. Other environmental conditions, such as nutrient and light availability can also provide strong bottom-up constraints in plankton communities^{12,13} and are particularly important along coastal boundaries¹⁴. Within the SCC, coastal upwelling creates strong spatial gradients in temperature, nutrients, and light^{15,16} (Supplementary Data Fig. 1). Previous studies have shown that phytoplankton and zooplankton communities vary along these gradients¹⁷⁻¹⁹. Furthermore, changes in seasonal nearshore upwelling are thought to drive distinct differences in phytoplankton and zooplankton assemblages across the

region with variation occurring on seasonal, interannual (El Niño/La Niña), and multidecadal (Pacific Decadal Oscillation) time frames^{20,21}. Within the microbial community however, the bulk of knowledge exists at a broad level of taxonomic and or functional groups, masking the effects of environmental perturbation within these broad groups and completely missing “cryptic” groups that cannot be identified with more traditional methods (such as bacterial and archaeal groups).

Metabarcoding and metagenomic datasets provide a crucial next step with which to explore the patterns and processes of marine microbial communities at a far higher resolution and in doing so, illuminate the key processes that structure the base of marine food web. However, our current understanding of the high taxonomic resolution, spatial patterns in microbial community structure and biodiversity are limited by the spatial and or temporal scale of sampling. Studies often focus on changes across space or time but rarely both²²⁻²⁴. Global datasets of marine microbiome data capture spatially extensive physical and ecological domains^{5,8,25} and can identify the large environmental gradients such as temperature that appear to shape communities across large ocean basins. In contrast, investigations conducted at singular stations identify changes in the marine microbiome through time²⁶⁻²⁹, exploring questions such as how succession within one group (such as phytoplankton) can drive changes in the overall community composition³⁰. However, the biotic and abiotic mechanisms that shape biodiversity and community composition patterns often remain uncertain⁴. Combined spatial and temporal metagenomic and metabarcoding sampling of marine microbial communities is necessary to illuminate the gaps in spatially or temporally explicit microbiome studies, such as whether trends happening in one location occur elsewhere or whether observed spatial patterns are conserved or vary across time.”

ii.) The characterization of communities as nearshore/off-shore types is not particularly meaningful, as their position from shore doesn't convey useful information about community function. Could these different communities also be called oligotrophic (for off-shore) and eutrophic (for nearshore) communities given the statement in the abstract describing the transition as “eutrophic nearshore to the oligotrophic offshore”? If so, this would be much more insightful than describing these communities as nearshore/offshore, which doesn't have much biological meaning other than position from the coast. This then highlights the idea that typical eutrophic organisms are found with more upwelling, which again either supports what is known about upwelling regions, or a better description needs to be provided about how it is similar/different from what is known from previous work.

Nearshore and offshore community clusters were an emergent property of our SOMs generated directly from ASV data. While it is true that on average, the nearshore coastal environment is meso/eutrophic in contrast to the oligotrophic offshore, we feel that defining these communities by this relationship is incorrect.

The nearshore is a highly variable environment and communities that occur in this region experience a wide range of productivity conditions.

However, we agree with the reviewer that this relationship should be better explained and have done so with the addition of the below text which follows our initial description of the SOM output (Lines 90-95), “For the five key functional groups, these clusters aligned with waters of contrasting trophic status. On average, stations found in the northeast, nearshore corner of the sampling grid experienced mesotrophic ($2.5\text{-}8\ \mu\text{g Chl-a L}^{-1}$) and eutrophic conditions ($> 8\ \mu\text{g Chl-a L}^{-1}$)⁴⁰. This contrast strongly with the oligotrophic conditions found in most of the stations further offshore, where chlorophyll is typically low ($< 2.5\ \mu\text{g Chl-a L}^{-1}$) (Supplementary Data Fig. 1i).”

We also estimated the relative rate of export (ef-ratio) for samples with primary productivity data. The ef-ratio is defined as $\text{new production}/\text{total production} = \text{export production}/\text{total production}$. Higher ef-ratio values indicate higher rates of export from the surface ocean (indicating higher rates of carbon sequestration). With this value we highlight the ecological function associated with SOM clusters in a new supplementary figure (Supplementary Data Figure 4, shown below). We find a strong relationship between the mean ef-ratio at stations and the relative frequency at which the nearshore cluster is observed at a station for both cyanobacteria and photosynthetic eukaryotic protists.

Manuscript text (Lines 112-121): “The export rate of primary production (ef-ratio) also varied in relation to SOM clusters (Supplementary Data Fig. 4). Here ef-ratio is defined as $\text{new production}/\text{total production} = \text{export production}/\text{total production}$ ⁴², where higher ef-ratio values indicate increased export of surface primary productivity to depth (important for carbon sequestration within the ocean). This was particularly evident in both the cyanobacteria and photosynthetic eukaryotic protists SOM clusters (Supplementary Data Fig. 4c-d), which both showed strong and significant relationships between the frequency with which their nearshore cluster was observed at a given station and the mean ef-ratio at that station over the seven years. The strong link between ef-ratio and proportion of nearshore and offshore communities highlights the connection between community structure and function, in this case the export of carbon from the ocean surface.”

Supplementary Data Fig. 4: Relationship between ef-ratio (calculated from Eq. 2 of Laws et al. 2011⁹⁴) and the frequency of SOM clusters per station from 2014-2020. Cragg and Uhler's pseudo R^2 was used to assess the goodness of fit⁹⁵ between mean ef-ratio and frequency. The shading represents a 95% confidence interval around each model fit.

Along these lines, could the authors also create a schematic to illustrate the relationship between the nearshore/offshore nitracline slope, up-welling and the community composition to facilitate understanding of how this change influences community composition and function? It might be particularly compelling if the authors adopt the oligotrophic/eutrophic communities nomenclature and show the steeper nitracline slope created from stronger nutrient upwelling leads to a larger proportion of eutrophic(nearshore) community members overall. This would make it easier for readers to understand the relationship between these different components of their work. While Extended data Fig. 13 is helpful to outline the relationships between BEUTI, nitracline slope and spatial relationship with distance from shore, a schematic about how this impacts the microbial community would be helpful.

We updated Figure 6 to include two illustrative examples of how changes in the regional slope of the nitracline can drive changes in the relative dominance of community types (nearshore or offshore). We decided to make these illustrative, data-driven examples rather than a schematic as we feel that these examples directly connect our description of regional nitracline slope (originally Suppl. Data Fig. 13, now Supplementary Data Fig. 14) to our results in Figure 6c-g, highlighting how the regional slope of the nitracline, and by extension the regional supply of nutrients to the surface ocean drives changes in regional community structure.

Fig. 6: Temporal shifts in regional nitracline gradients align with relative community dominance. a, Illustrative example highlighting a cruise (Winter 2015) where the regional slope in nitracline depth is relatively low. The black line indicates the regional slope in the nitracline depth. Points indicate individual samples taken during this cruise. The color of the points indicates cyanobacteria communities that were identified by SOMs as either “nearshore” (blue) or “offshore” (red). **b,** Illustrative example highlighting a cruise (Summer 2019) where the regional slope in nitracline depth is much greater. The black line indicates the regional slope in the nitracline depth. Points indicate individual samples taken during this cruise. The color of the points indicates cyanobacteria communities that were identified by SOMs as either “nearshore” (blue) or “offshore” (red). **c-g,** Proportion of samples per cruise that were identified by SOMs as “nearshore” communities relative to the slope in the nitracline across the entire region. Shapes represent the different seasons during which cruises took place (circle = winter, square = spring, diamond = summer, triangle = fall) and the colors represent samples that were collected from 2014-2016 (red), 2017-2018 (blue), or 2019-2020 (gold). Data were fitted as separate linear models per phase, shading represents the 95% confidence interval around the model fit.

Fig. 3 How does the nearshore community make up 100% of the sample when the nitracline depth is largest (90m) since nitracline depth increases with distance from shore? Why did this figure change completely from the original form? This is confusing and should be explained in either the legend or the text.

We thank the reviewer for noting this error (nearshore/offshore were mislabeled in the legend), the figure has been corrected and is shown below.

Reviewers' Comments:

Reviewer #3:

Remarks to the Author:

The authors have sufficiently addressed my concerns and improved the clarity and presentation of the their work with their revisions.

Response to Reviewers (3rd Round)

Reviewer #3 (Remarks to the Author):

The authors have sufficiently addressed my concerns and improved the clarity and presentation of the their work with their revisions.

We thank the reviewers for their diligence and thoughtfulness throughout the entire review process. We believe that their combined insight and attention to detail greatly improved this manuscript from its initial submission.